# SGLT2 Inhibitors as the Most Promising Influencers on the Outcome of Non-Alcoholic Fatty Liver Disease

**DOI:** 10.3390/ijms23073668

**Published:** 2022-03-27

**Authors:** Luigi Mirarchi, Simona Amodeo, Roberto Citarrella, Anna Licata, Maurizio Soresi, Lydia Giannitrapani

**Affiliations:** 1Department of Health Promotion Sciences, Maternal and Infant Care, Internal Medicine and Medical Specialties (PROMISE), University of Palermo, 90127 Palermo, Italy; mirarchi.luigi@tiscali.it (L.M.); simona.amodeo@unipa.it (S.A.); roberto.citarrella@unipa.it (R.C.); anna.licata@unipa.it (A.L.); maurizio.soresi@unipa.it (M.S.); 2Institute for Biomedical Research and Innovation (IRIB), National Research Council, 90146 Palermo, Italy

**Keywords:** NAFLD, metabolic syndrome, type 2 diabetes mellitus, SGLT2

## Abstract

Non-alcoholic fatty liver disease (NAFLD), the most frequent liver disease in the Western world, is a common hepatic manifestation of metabolic syndrome (MetS). A specific cure has not yet been identified, and its treatment is currently based on risk factor therapy. Given that the initial accumulation of triglycerides in the liver parenchyma, in the presence of inflammatory processes, mitochondrial dysfunction, lipotoxicity, glucotoxicity, and oxidative stress, can evolve into non-alcoholic steatohepatitis (NASH). The main goal is to identify the factors contributing to this evolution because, once established, untreated NASH can progress through fibrosis to cirrhosis and, ultimately, be complicated by hepatocellular carcinoma (HCC). Several drugs have been tested in clinical trials for use as specific therapy for NAFLD; most of them are molecules used to cure type 2 diabetes mellitus (T2DM), which is one of the main risk factors for NAFLD. Among the most studied is pioglitazone, either alone or in combination with vitamin E, glucagon-like peptide-1 (GLP-1) receptor agonists, dipeptidyl peptidase-4 (DPP-4) inhibitors. Actually, the most promising category seems to be sodium-glucose cotransporter (SGLT2) inhibitors. Their action is carried out by inhibiting glucose reabsorption in the proximal renal tubule, leading to its increased excretion in urine and decreased levels in plasma. Experimental studies in animal models have suggested that SGLT2 inhibitors may have beneficial modulatory effects on NAFLD/NASH, and several trials in patients have proven their beneficial effects on liver enzymes, BMI, blood lipids, blood glucose, and insulin resistance in NAFLD patients, thus creating strong expectations for their possible use in preventing the evolution of liver damage in these patients. We will review the main pathogenetic mechanisms, diagnostic modalities, and recent therapies of NAFLD, with particular attention to the use of SGLT2 inhibitors.

## 1. Introduction

Non-alcoholic fatty liver disease (NAFLD) is a condition characterized by the presence of an excessive accumulation of fat in the liver, which can be classified histologically as non-alcoholic fatty liver (NAFL) or non-alcoholic steatohepatitis (NASH). The defining feature of NAFL in histology is the presence of at least 5% of lipids within hepatic cells, in the absence of evidence of hepatocellular damage [1], which is not linked to alcohol or drug abuse or other conditions that can cause liver steatosis. NASH is also defined as the presence of at least 5% of lipids within hepatic cells but in association with signs of inflammation and hepatocellular damage, with or without fibrosis, and which can progress toward liver cirrhosis, liver failure, or the development of hepatocellular carcinoma (HCC) [2].

To diagnose NAFLD, excessive alcohol consumption should be excluded both in terms of g/day (>24 g/day for men and >16 g/day for women) and number of drinks/day (≥5 for men and ≥4 for women); an intake higher than the quantities indicated would, in fact, configure the diagnosis as alcoholic liver disease (ALD) [1]. It is also necessary to evaluate and exclude the presence of secondary causes of liver steatosis, such as drugs, hereditary metabolic disorders, and other chronic liver diseases [3].

Due to the recent spread of obesity and metabolic syndrome (MetS), NAFLD is now considered to be the chronic liver disease with the highest rate of increase globally, with the overall prevalence of NAFLD being currently estimated at 25% in the general adult population [4].

Moreover, in the attempt to overcome the current NAFLD nomenclature and underline a “positive” definition, the name metabolic dysfunction-associated fatty liver disease (MAFLD) has recently been proposed, using metabolic dysfunctions as diagnostic criteria without excluding other causes of other chronic liver diseases [5].

Although widespread across many continents, the prevalence of NAFLD is extremely variable globally, with the highest prevalence rates reported in South America (31%), the Middle East (32%), the USA (24%), and Europe (23%), with peaks in Greece and Spain (40%), while the lowest rates are reported in Africa (14%) [6].

NAFLD is the most common liver disease in the Western world, where it affects 30%–40% of men and 15%–20% of women, and it exceeds 70% in patients with type 2 diabetes mellitus (T2DM) [7].

The degree and prevalence of liver fibrosis secondary to NAFLD also tend to progressively increase with age, being an expression of a longer duration of the disease and longer exposure to liver damage factors [8].

Over the past 20 years, NAFLD and its stages have represented the fastest growing indication for liver transplantation, as reported by European Liver Transplant Registry (ELTR) data.

In the USA in 2018, it was the second-leading cause of transplantation, with 21.5% of orthoptic liver transplant (OLT) cases, and an equally significant increase has occurred in Europe, where post-NASH cirrhosis was indicated for transplantation in 8.4% of cases in the same year [9].

## 2. Risk Factors

The main risk factors for the development of the disease are attributable to the state of insulin resistance and, therefore, to MetS (visceral obesity, T2DM, dyslipidemia, arterial hypertension), which is its phenotypic expression [6].

The diseases characterizing MetS are not only highly prevalent in patients with NAFLD, but the presence of one or more of these conditions increases the risk of developing NAFLD itself [10].

This bi-directional association between NAFLD and the components of MetS has been strongly consolidated, so much so that NAFLD is considered the hepatic manifestation of MetS, and it commonly coexists with T2DM, which is a risk factor for its progression to fibrosis, cirrhosis, and cancer [11,12].

Therefore, in consideration of the epidemic-level increase in obesity and T2DM (diabesity), as well as the scarcity of truly effective and specific treatments for liver steatosis, the prevalence of this disease is destined to increase [13].

Moreover, alongside the well-known classic metabolic risk factors, some others are emerging for which a correlation with NAFLD has been observed: hypothyroidism, sleep apnea syndrome, hypopituitarism, hypogonadism, pancreatic-duodenal resection, psoriasis, osteoporosis, chronic obstructive pulmonary disease (COPD), polycystic ovary syndrome (PCOS), atherosclerosis, chronic renal failure, extrahepatic neoplasms, hyperuricemia [3].

## 3. Etiology and Pathophysiology

NAFLD has a multifactorial etiology that arises from a complex interaction between genetic and environmental factors.

Among the genetic factors, there are both specific genetic polymorphisms (patatin-like phospholipase domain-containing protein 3 (PNPLA3) gene, transmembrane 6 superfamily member 2 (TM6S2) gene) and epigenetic modifications [14].

Among the environmental factors are excess caloric intake, absence of physical activity, adipokine dysregulation, endoplasmic reticulum stress, oxidative stress, lipotoxicity, intestinal microbiota dysbiosis, and endocrine alterations, in addition to the aforementioned obesity and insulin resistance.

The classic pathogenetic model of NAFLD is described as the “two-hits hypothesis”. According to this theory, the interaction of several predisposing factors determines the impact on the liver parenchyma initially through the accumulation of triglycerides (first hit), and this accumulation acts as a sensitizing factor. If steatosis is not promptly countered, an inflammatory process can develop, with mediators of inflammation (especially cytokines and adiponectins), mitochondrial dysfunction, lipotoxicity, glucotoxicity and oxidative stress causing the second hit. These secondary effects can lead to steatohepatitis and fibrosis [15]. So, progression from NAFL to NASH occurs when adaptive mechanisms are no longer able to protect hepatocytes from lipotoxicity, causing inflammation, oxidative stress, and apoptosis.

More recently, this “two hits” theory has been transformed into the “multiple hits” theory. In fact, Tilg and Moschen have re-evaluated the previous theory, proposing a new model in which multiple aetiological agents interact simultaneously in determining liver damage. According to this new model, steatohepatitis and liver steatosis represent two distinct pictures of the disease. Several extrahepatic factors (intestinal microbiota, adipose tissue, genetic polymorphisms) can play a pathogenetic role in the development of NASH, regardless of the presence of NAFLD [16]. However, not all people with NAFLD progress to NASH, so the main goal of the research in this field is to identify the factors involved in NASH development because, once established, if left untreated, it risks progressing through fibrosis, cirrhosis, and, ultimately, being complicated by HCC.

From the metabolic point of view, hepatocytes appear to have a function similar to that of adipose tissue cells. In fact, when adipocytes reduce their ability to store energy in excess (as occurs in obese patients), this is deposited in the form of triglycerides. The increase in circulating fatty acids (free fatty acids—FFAs), deriving from accelerated lipolysis and the increase in fatty acid storage in adipose tissue, leads to the accumulation of ectopic fat (liver, skeletal muscle).

In addition, the prolonged exposure of hepatocytes to elevated lipid and carbohydrate levels causes lipotoxicity and glucotoxicity, respectively, with the development of steatosis and progression to NASH. The inability of hepatocytes to dispose of excess FFAs generates lipoapoptosis, the process underlying the development of steatohepatitis [17]. The inflammatory process generated by the activation of Kupffer, dendritic and hepatic stellate cells (HSCs) is amplified through the secretion of proinflammatory cytokines (IL-6, IL-17a) [18], determining the activation and subsequent infiltration of the liver by neutrophils, monocytes, natural killer cells, T lymphocytes and especially macrophages. Under physiological conditions, this process tends to stop the accumulation of lipids by eliminating the infiltrated hepatocytes; however, if the compensatory capacity fails due to the activation of HSCs, fibrotic tissue would replace the liver parenchyma. HSCs, generally quiescent, respond to the inflammatory stimulus as well as to other profibrotic signals (TGF-β, leptin, PDGF) and, once activated, proliferate and differentiate in myofibroblasts, which result in extracellular matrix deposition causing fibrosis [19,20]. Endocrine factors secreted by adipose tissue such as adiponectin and leptin may contribute to the liver damage typical of NAFL in obese patients; the increase in weight, in fact, determines a dysregulation of adipokines, which take on a more steatogenic, inflammatory, and fibrogenic profile [15]. Furthermore, in obese subjects, the infiltration of immune cells into the adipose tissue stimulates the production of classic proinflammatory cytokines, which, in turn, interact with adipokines, thus predisposing them to the development of insulin resistance [21,22].

Autophagy has been emerging in recent years as an important issue related to liver diseases. It is a catabolic mechanism of the cell, which plays a fundamental role in organ homeostasis, immune response, and energy balance. The modulation of autophagy, through its anti-steatogenic and anti-inflammatory properties and the protective properties of mitophagy (a selective form of autophagy for the sequestration of damaged mitochondria) on hepatocytes may represent a selective therapeutic target in NAFLD. In addition, insulin resistance, oxidative stress, hyperglycemia, and lipotoxicity can lead to a reduction in autophagy processes and contribute to the pathogenesis of NAFLD [23].

In fact, in experimental mouse models fed with high lipid content diets, it has been shown that the deletion of parts of the autophagic pathway in liver cells led to an increased accumulation of lipid droplets, as well as endoplasmic reticulum stress, increased hepatocyte damage, and increased production of proinflammatory cytokines [24,25]. The reduction in autophagic processes can be related to the different phases of NAFLD, as evidenced in hepatic endothelial cells in patients with NASH compared to patients with normal liver or simple steatosis [26]. Although the data must be corroborated, it is paradigmatic that the main measures aimed at containing and countering the progression of NAFLD, such as calorie restriction, exercise, use of metformin, GLP-1 receptor agonists, SGLT2 inhibitors, act on important autophagy targets [27,28,29,30] (Figure 1). Ezetimibe, although not yet approved for the treatment of NAFLD, also appears to target the same pathways for activating autophagy.

## 4. Diagnosis

NAFLD includes a broad spectrum of hepatic alterations, which mostly run asymptomatically except in the most advanced stages, and in most cases, the diagnosis is made in patients undergoing screening because they are suffering from MetS.

Although serum aminotransferase (AST/ALT) levels are generally used in the surveillance of patients with suspected NAFLD, they are rarely altered and, indeed, in 80% of NAFLD cases, they are not altered at all [31,32].

Furthermore, reduced ALT levels can be found in patients in the most advanced stages of the disease [33,34] and do not seem to be correlated with the histological picture [35]. In addition, elevated serum ferritin levels may be found in patients with NAFLD, with levels > 1.5 times normal mostly related to a more advanced stage of fibrosis [36].

The most used imaging technique for diagnosing NAFLD remains ultrasound (US) due to the simplicity of its use, ease of access, wide dissemination, and low cost.

It also has the ability to provide additional information on liver conditions and can be supplemented by specific complementary assessments, such as mesenteric fat assessment [37,38]. It also has suitable efficacy in the diagnosis of moderate and severe steatosis, with 74%–91% sensitivity and 85%–98% specificity, depending on the case series [39]. In fact, some studies have confirmed the diagnostic sensitivity of US even in moderate steatosis using magnetic resonance spectroscopy as a reference standard, a highly sensitive method for identifying even mild forms of steatosis [40]. However, US has a limited ability to identify mild steatosis since it has low sensitivity when the triglyceride content is less than 20% [41]. It is possible to make a semi-quantitative estimate of steatosis severity as a function of the degree of attenuation of the US beam at depth. Quantitative US parameters include the attenuation coefficient (AC) and the backscatter coefficient (BSC), which have been developed for the quantification of liver fat. The AC measures the loss of ultrasonic energy in the tissue and provides a quantitative parameter analogous to the qualitative loss of vision of the deeper structures observed in the fatty liver. The BSC measures the energy derived from the return of the US beam that encounters the tissue and provides a parameter similar to echogenicity [42].

With US, steatosis is diagnosed by the presence of small and high-intensity echoes that give the liver parenchyma a hyperechoic appearance compared to the renal cortex, which is called “bright liver”. In addition, other parameters such as the appearance of intrahepatic vessels and the diaphragm can also be taken into consideration to make the diagnosis [43]. From a US point of view, it is, therefore, possible to classify steatosis as follows: absent, when the liver parenchyma does not show US changes; mild, when the liver parenchyma appears diffusely hyperechoic with normal visualization of the vessels, particularly of the portal vein walls, and normal visualization of the diaphragm; moderate, when there is a marked increase in liver echogenicity and a simultaneous reduction in visualization of the portal vein wall and diaphragm; severe, when there is a marked increase in liver echogenicity with reduction or failure to visualize the portal vein and diaphragm as well as the posterior part of the right lobe [44,45].

Although quantitative parameters have shown the potential for the accurate assessment of fatty liver disease, US devices and operator variables need to be considered to further verify accuracy, reproducibility, and repeatability [46,47]. To these aspects is to be added the poor diagnostic capacity in obese subjects since the sensitivity and specificity in B-mode decrease with the increase in body mass index (BMI) and vary between 49%–100% and 75%–95%, respectively [48].

Furthermore, it must be remembered that US is not able to detect the presence of NASH or its progression to liver fibrosis unless there are already signs of cirrhosis and/or portal hypertension.

More recently, the evaluation of the controlled attenuation parameter (CAP) has been introduced. CAP is a measure of the acoustic attenuation of ultrasounds as they pass through the liver parenchyma, and it is measured in decibels/meter. It is an application of Fibroscan, which is acquired simultaneously with the measurement of liver stiffness and, since its value correlates with the degree of steatosis, it provides a quantitative measurement of steatosis itself [38]. A strong correlation between CAP and liver fat content at biopsy has been demonstrated [49,50]. The literature reports different cut-off values for CAP in relation to the different degrees of steatosis defined at histological examination, from S0, which indicates the absence of steatosis, to S3, which indicates the highest degree of steatosis. However, there is a failure rate in NAFLD patients related to the physical features of the patients; that is, the probability of failure increases in patients with a BMI > 30 or in the presence of an increase in waist circumference because it interferes with impulse transmission. There are two types of probes for performing the examination: probe M and probe XL, with the latter providing more reliable data in obese patients [51]. Furthermore, a recent study performed on 40 patients compared the data obtained from CAP to the histological data, confirming that CAP has a specificity of 87.5% in establishing the presence of steatosis, although the results on the reliability in establishing the degree of steatosis are controversial. In fact, it would seem that, using the XL probe compared to the M probe, CAP values tend to be higher [52].

For the evaluation of liver fibrosis, new noninvasive imaging methods of the “shearwave” elastographic type have become widespread, which permits an evaluation of the elastic and mechanical properties of the tissues to be carried out.

More recently, both the EASL and AISF guidelines suggest that the combined association between elastography and serum markers, such as Fibrosis-4 index (Fib-4) and NAFLD fibrosis score (NFS), allows the accurate definition of the risk of fibrosis and may be useful in selecting subjects to be biopsied [1,2].

## 5. Therapy

Specific treatment for NAFLD does not yet exist and presents a difficult challenge; if one were to be found, it would not only reduce the risk of liver disease progression but also cardiovascular risk [53]. In non-diabetic subjects to date, the only weapons available are lifestyle interventions, such as physical activity and calorie restriction, which are often difficult to carry out and maintain. Vitamin E administration has been shown to protect from progression toward more severe forms, such as fibrosis and cirrhosis, in these patients [54].

Among the diets, the one that seems to have the greatest impact is the Mediterranean diet (MD), a diet low in saturated fats and animal proteins, rich in antioxidants and fiber, and with an adequate ratio between omega-6 and omega-3 [55]. The MD is based on substances such as polyphenols, vitamins, and other compounds with anti-inflammatory and antioxidant action. In particular, polyphenols seem to have different hepato-protective activities. They are divided into flavonoid and non-flavonoid polyphenols [56]. In addition to influencing the flavor and color of certain foods (e.g., fruit and vegetables), flavonoids have an anti-inflammatory and antioxidant action [57,58,59]. Among the non-flavonoids, resveratrol performs a hepato-protective function by interacting with the homeostasis of the vessels, the function of platelets, and the coagulation system [60].

Vitamins are also an important element on which the MD is based, as they are able to reduce cellular stress and thus prevent the progression of NAFLD. Among the vitamins, vitamin D exhibits immunomodulatory, anti-inflammatory, and insulin-sensitizing properties [61]. In addition to vitamin D, vitamin C, thanks to its antioxidant properties, also has a protective function on the liver parenchyma [62,63].

However, in patients with T2DM and NAFLD, where the risk of the progression of liver damage is greater, vitamin E alone does not significantly modify the progression toward fibrosis, while the therapy of a combination of vitamin E with pioglitazone appears to improve the histological picture when compared to placebo [64]. On the other hand, while pioglitazone seems to show a statistically significant improvement compared to placebo, the side effects of its use are substantial, including water retention, weight gain, and cardiovascular risk, thus limiting its use [65].

A possible role in reducing the accumulation of fat in the liver in morbidly obese patients appears to be played by the inhibitor of proprotein convertase subtilisin/kexin 9 (PCSK9).

PCSK9 is a protein capable of binding and degrading the LDL cholesterol receptor (LDL-C), thus preventing its internalization and degradation at the cellular level, with a consequent increase in circulating LDL-C levels. Therefore, the inhibition of PCSK9 seems to be able to reduce the accumulation of fat in the liver, but not the risk of progression to fibrosis [66].

Another class of drugs that could be used is the glucagon-like peptide-1 (GLP-1). Physiologically, GLP-1 produced by the small intestine and proximal colon L cells regulates plasma glucose levels by stimulating the secretion of insulin by the pancreas and inhibiting that of glucagon in a glucose-dependent manner; it can improve insulin resistance, favoring weight loss through delayed gastric emptying and, consequently, a reduction in appetite [53]. GLP-1 is degraded by dipeptidyl peptidase-4 (DPP-4), which is highly expressed in the liver [67]. Elevated circulating DPP-4 levels appear to be associated with the severity of liver disease in patients with NAFLD [68]. Therefore, it is plausible to hypothesize that DPP-4 inhibitors could improve the histological characteristics of NAFLD and NASH [69]. Given this, GLP-1 may have beneficial effects on NAFLD both indirectly, through weight loss, improving insulin resistance, and controlling blood sugar (1), but also by acting directly on hepatocytes, reducing the triglycerides they contain [70,71,72]. New molecules that, while certainly needing further studies, have already demonstrated promising efficacy include tirzepatide. This drug acts on both the glucose-dependent insulinotropic polypeptide (GIP) and the GLP-1 receptor. The use of this molecule in a phase 2 randomized trial of patients with T2DM demonstrated a significant reduction in NASH-related biomarkers (AST/ALT, keratin 18, fragment M30, procollagen III) and an increase in adiponectin levels, which has antifibrotic and antisteatogenic activity in the liver [73].

## 6. SGLT2 Inhibitors

Among all the drugs used in T2DM therapy, one seems to be more promising and shows evident benefits in patients with NAFLD: sodium-glucose cotransporter (SGLT2) inhibitors. In normo-glycaemic conditions and with preserved renal function, the renal filtration threshold of glucose is equal to 180 g/day; when plasma glucose levels exceed this threshold, it is eliminated in the urine together with sodium. There are two types of cotransporters: SGLT1, present both in the intestine and below the Bowman’s capsule in the thick portion of the proximal convoluted tubule, and SGLT2, which instead is present exclusively in the renal tubule. Approximately 97% of the glucose is reabsorbed upstream of the proximal tubule by SGLT2, while the remaining glucose is reabsorbed downstream by SGLT1. In hyperglycaemic states, the kidneys increase their renal resorption capacity up to a maximum of 600 g/day to prevent renal loss of glucose [74,75]. This effect, mediated by the SGLT cotransporter, not only leads to an increase in the reabsorption of glucose but also of sodium and liquids so that the inhibition of this transporter would not only have beneficial effects on glycaemic control but also in sodium homeostasis and water retention. SGLT2 inhibitors were developed for the management of T2DM, inhibiting the reabsorption of glucose in the kidney and decreasing blood glucose levels. Several recent clinical studies have shown that they can improve liver function in patients with NAFLD as well as in T2DM (Figure 2). Moreover, for this consistent impact on natriuresis and diuresis, they have been proposed for the co-management of T2DM and refractory ascites in cirrhosis [76]. For these pathophysiological reasons, SGLT2 inhibitors are important, promising therapeutic agents in patients with NAFLD [71,72].

Several studies have shown that the administration of SGLT2 inhibitors has proven efficacy in terms of reducing plasma levels of ALT, body weight [77], and blood pressure and improving glycated hemoglobin (HbA1c) [78], reducing risks of cardiovascular and renal diseases, all of which are factors that inhibit the progression to NASH [79]. In addition, by reducing fat mass, SGLT2 inhibitors prevent the release of inflammatory cytokines by adipocytes, thus decreasing the inflammatory effect, which is one of the main causes of NASH progression [80].

To date, different studies have evaluated the efficacy of SGLT2 inhibitors in the treatment of NAFLD [81] (Table 1).

In 2018, a randomized controlled clinical trial (the E-LIFT trial) conducted in India on 50 patients with T2DM and NAFLD showed that the addition of empagliflozin 10 mg to standard T2DM therapy for 20 weeks resulted in a significant reduction (16.2% vs. 11.3%) in liver fat at the end of treatment, as measured by proton density MRI, and an improvement in serum ALT levels [82].

Another trial conducted by Kahl et al. on T2DM patients, 80% with NAFLD, compared a 25 mg daily dose of empagliflozin with placebo demonstrated a significant reduction in liver fat content (LFC) using magnetic resonance spectroscopy [83].

In addition, Shimizu et al. conducted a 24-week open-label controlled clinical trial of 57 patients with T2DM and NAFLD, randomized into a dapagliflozin group (5 mg/day; *n* = 33) or a control group (*n* = 24). Fatty liver and liver fibrosis were assessed using transient elastography to measure CAP and liver stiffness, respectively. This study demonstrated a significant reduction in CAP in the dapagliflozin group, as well as a reduction in liver stiffness [84].

Ito et al. compared 50 mg of ipragliflozin versus pioglitazone in addition to standard care, having as a primary outcome a change from baseline in the liver-to-spleen attenuation ratio (L/S ratio) on computed tomography at week 24. This study observed an improvement in liver steatosis assessed using the L/S ratio, reduced serum aminotransferase levels, and beneficial effects on glycaemic parameters. Compared to pioglitazone, there were significant decreases in body weight and visceral fat [85].

A study by Ohki proved the efficacy of ipragliflozin in a group of patients not responsive to incretin and DPP4-I therapy in improving glycaemic control, reducing body weight, and transaminase levels [86].

Han et al. conducted a study comparing the effects of 50 mg of ipragliflozin in patients with T2DM and NAFLD who were already receiving pioglitazone and metformin versus patients with pioglitazone and metformin alone. The results were evaluated by the fatty liver index and the NAFLD liver fat score, with a significant reduction in patients treated with SGLT2, although this difference between the two groups was not demonstrated for CAP [87].

In a prospective study, nine patients with NAFLD complicated by T2DM were treated with a daily 100 mg dose of canagliflozin for 24 weeks and were evaluated by liver histology at baseline and at 24 weeks after the initiation of therapy. The main primary endpoint was a histological improvement, defined as a decrease in NAFLD activity score by one point or more without worsening the stage of fibrosis. All nine patients achieved histological improvement. Six patients showed an improvement in insulin resistance, and further three patients showed a partial improvement in insulin secretion function [88].

In T2DM patients with NAFLD, Inoue et al. evaluated the effect of 100 mg of canagliflozin administered once daily for one year on serological markers, body composition measured by bioelectrical impedance analysis, and liver fat by MRI. A significant reduction in body and fat mass was shown at 6 and 12 months, without a significant reduction in muscle mass. The hepatic fat fraction was reduced from a baseline of 17.6% ± 7.5% to 12.0% ± 4.6% after 6 months and 12.1% ± 6.1% after 12 months, while serum liver enzymes and type IV collagen concentrations improved. From a mean baseline HbA1c of 8.7% ± 1.4%, canagliflozin significantly reduced HbA1c after 6 and 12 months to 7.3% ± 0.6% and 7.7% ± 0.7%, respectively (*p* < 0.0005 and *p* < 0.01) [89].

A further study conducted by Nishimiya et al. evaluated the efficacy of canagliflozin at a dosage of 100 mg once a day in a group of 10 patients with T2DM and NAFLD in addition to the therapy practiced. The degree of steatosis was assessed using three different imaging methods: MRI, computed tomography (CT), and transient elastography. The bio-humoral parameters of glycemic, lipid, and liver function were evaluated. The six-month study confirmed the efficacy of canagliflozin in improving HS, insulin resistance, reduction in adipose tissue, and inflammation [90]. In addition, Goutam et al. evaluated the role of canagliflozin at a dosage of 100 mg/day in reducing body weight, glycated hemoglobin, and improving liver function tests [91].

Sumida et al. (LEAD trial) and Shibuya et al. in their studies demonstrated that the use of luseogliflozin 2.5 mg/die in T2DM and NAFLD patients led to the improvement of several metabolic and liver function-related parameters and reduced fat [92,93].

Finally, Seko et al. published a retrospective study to evaluate the efficacy of SGLT2-I in a group of patients with histologically proven NAFLD and T2DM treated for 24 weeks. A total of 24 patients had received SGLT2-I (canagliflozin 100 mg/day or ipragliflozin 50 mg/day). While 21 patients had been treated with DPP4-I (sitagliptin 100 mg/day). In addition, in this retrospective study, the results encourage the use of SGLT2-I in patients with T2DM and NAFLD, demonstrating a significant reduction in weight and glycated hemoglobin. Transaminase activity was similarly reduced between the two groups [94].

**Table 1 ijms-23-03668-t001:** Features of the studies published on SGLT2-I effects on NAFLD patients.

Author/Year	Design/Duration	Drug	Posology	Outcome(s)	Conclusion
Ohki T, et al., [86] 2016;	Observational (retrospective) 45 weeks median Patients n° 24	Ipragliflozin	Ipragliflozin 50 mg + DPP-4I (n° 13) vs. Ipragliflozin 50 mg + GLP-1 RA (n° 11)	Changes ALT levels and body weight at the end of the follow-up	Ipragliflozin normalizes ALT levels and improves glycemic control, it reduces body weight, FIB-4 score, in patients who did not respond to incretin-based therapy
Seko Y, et al., [94] 2016;	Observational (retrospective) 24 weeks Patients n° 45	DPP4-I vs. SGLT2-I	Sitagliptin 100 mg daily (n° 21) vs. Canagliflozin 100 mg daily or ipragliflozin 50 mg daily (n° 24)	Correlation between changes in aminotransferase, body weight, glycemic control, and HbA1c	The reductions in ALT and HbA1c were similar between SGLT2-I and DPP4-I groups, whereas body weight was significantly reduced in the SGLT2-I group compared with the DPP4-I group
Ito D, et al., [85] 2017;	RCT, OL, single center 24 weeks Patients n° 66	Ipragliflozin vs. Pioglitazone	Ipragliflozin 50 mg daily (n° 32) vs. Pioglitazone 15–30 mg daily (n° 34)	Change from baseline in L/S ratio on CT	Both had benefits on NAFLD and glycemic control; Ipragliflozin reduced body weight and abdominal fat area
Kuchay MS, et al., [82] 2018; E-LIFT Trial	RCT, OL, single center. 20 weeks Patients n° 50	Empagliflozin vs. ST T2DM	Empagliflozin + ST T2DM (n° 25) vs. ST T2DM (n° 25)	Change in liver fat was measured by MRI-PDFF. Secondary outcome measures were change in ALT, AST, and GGT levels	Empagliflozin reduces liver fat and improves ALT levels in patients
Shimizu M, et al., [84] 2018;	RCT, OL, single center. 24 weeks Patients n° 57	Dapagliflozin vs. ST T2DM	Dapagliflozin 5 mg daily (n° 33) vs. ST T2DM (n° 24)	HS and fibrosis were assessed using transient elastography to measure CAP and liver stiffness, respectively	Dapagliflozin improves HS and attenuates liver fibrosis in patients with significant liver fibrosis
Gautam A, et al., [91] 2018;	Observational. 24 weeks Patients n° 31	Canagliflozin + ST T2DM	Canagliflozin 100 mg daily + ST T2DM	Improves LFT and HbA1c	Canagliflozin controls HbA1c and reduce weight in type 2 diabetes, and significantly improves LFT
Shibuya T, et al., [93] 2018;	RCT, OL, single center, prospective. 24 weeks Patients n° 32	Luseogliflozin vs. Metformin	Luseogliflozin 2.5 mg daily (n° 16) vs. Metformin 1500 mg daily (n° 16)	Change in L/S ratio obtained by CT	Luseogliflozin significantly reduces liver fat deposition compared to metformin
Sumida Y, et al., [92] 2019;	Prospective, 24 weeks Patients n° 40	Luseogliflozin + ST T2DM	Luseogliflozin 2.5 mg once daily + ST T2DM (without insulin)	Change in HbA1c and hepatic fat content from baseline. The secondary endpoints were the changes: routine liver biochemistries, blood pressure, lipid profiles, and hepatic fibrosis markers	Improves HbA1c, transaminase levels, and hepatic fat content
Akuta N, et al., [88] 2019;	Prospective, OL, single center. 24 weeks Patients n° 9	Canagliflozin	Canagliflozin 100 mg daily	Histological improvement, defined as a decrease in NAFLD activity score without worsening in fibrosis stage	All patients achieved histological improvement. Scores of steatosis, lobular inflammation, ballooning, and fibrosis stage decreased at 24 weeks
Inoue M, et al., [89] 2019;	Prospective, OL, single center. 48 weeks Patients n° 20	Canagliflozin + ST T2DM	Canagliflozin 100 mg daily + ST T2DM	Change in body composition measured by bioelectrical impedance analysis method and hepatic fat fraction measured by MRI	Canagliflozin reduced body mass, fat mass, and hepatic fat content without significantly reducing muscle mass
Kahl S, et al., [83] 2020;	RCT, prospective, multi center. 24 weeks Patient n° 84	Empagliflozin vs. Placebo	Empagliflozin 25 mg daily (n° 42) vs. Placebo (n° 42)	Change in liver fat content measured with MRI	Empagliflozion reduces hepatic fat with excellent glycemic control and short known disease duration
Han E, et al., [87] 2020;	RCT, OL, single center. 24 weeks Patient n° 44	Metformin + Pioglitazone + Ipragliflozin vs. Metformin + Pioglitazone	Ipragliflozin 50 mg daily (n° 29) + Metformin + Pioglitazone vs. Metformin + Pioglitazone (n° 19)	Change in HS measured by fatty liver index, NAFLD liver fat score, and CAP	Ipragliflozin improves liver steatosis and reduces excessive fat in euglycemic patients
Nishimiya N, et al., [90] 2021;	Prospective, single center 24 weeks Patient n° 9	Canagliflozin + ST T2DM	Canagliflozin 100 mg daily + ST T2DM	Change in HS assessed using the hepatic MRI-PDFF	Canagliflozin improved HS reducing adiposity, insulin resistance, inflammation, and skeletal muscle volume

ALT: alanine aminotransferase; AST: aspartate aminotransferase; CAP: controlled attenuated pressure; CT: computed tomography; DPP4-I: dipeptidyl peptidase-4 inhibitor; FIB-4: Fibrosis-4 score; GGT: gamma-glutamyl transferase; GLP1-RA: glucagon-like peptide 1 receptor agonist; HS: hepatic steatosis; LFT: liver function test; L/S ratio: liver-to-spleen attenuation ratio; MRI-PDFF: magnetic resonance imaging estimated proton density fat fraction; NAFLD: non-alcoholic fatty liver disease; OL: open label; RCT: randomized controlled trial; SGLT2-I: sodium glucose cotransporter-2 inhibitors; ST: standard treatment; T2DM: type 2 diabetes mellitus.

## 7. Conclusions

The natural history of the evolution of NAFLD/NASH and the scarcity, if not total absence, of specific treatments for this condition necessitates increased attention to the possible application of innovative T2DM therapies also for this disease. Of particular interest are SGLT2 inhibitors that have been proven to be efficient in reducing liver fat content, AST/ALT levels, and even liver stiffness in several trials, making this class of drugs one of the most promising future treatments for the specific indication of NAFLD. Moreover, most of the studies conducted so far are retrospective or conducted on a small number of patients; therefore, prospective, randomized, controlled studies are needed to deepen the impact of SGLT2 inhibitors both on the already known clinical-instrumental parameters and on newer genetic and epigenetic predictors of NAFLD evolution.

## Figures and Tables

**Figure 1 ijms-23-03668-f001:**
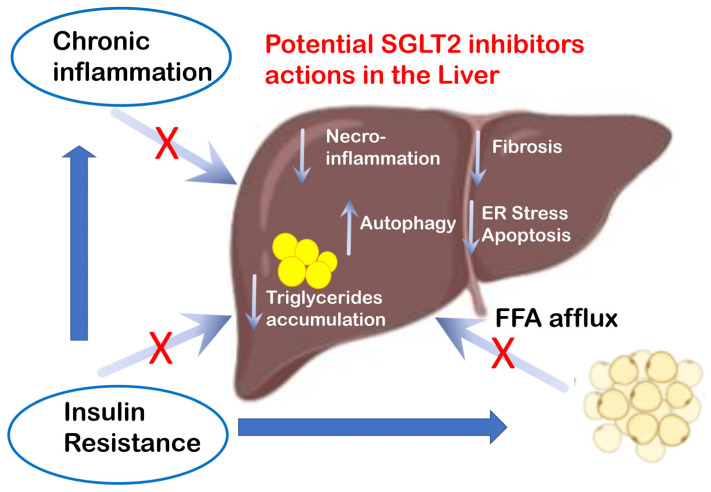
Potential actions of SGLT2-i on several functions related to NAFLD.

**Figure 2 ijms-23-03668-f002:**
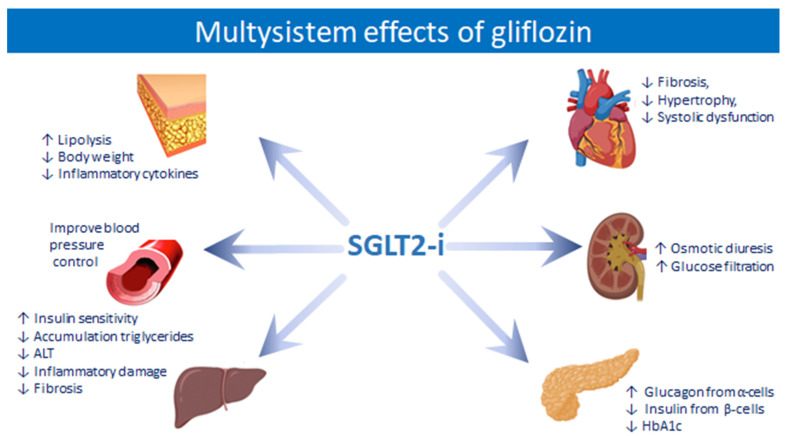
Effects of SGLT2-I on several organs and functions.

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
