# Peer review of "SGLT2 Inhibitors as the Most Promising Influencers on the Outcome of Non-Alcoholic Fatty Liver Disease"

_ijms, 2022, doi:10.3390/ijms23073668_

Round 1

Reviewer 1 Report

This manuscript by Giannitrapani et. al. provided the summary of SGLT2 inhibitors as the most promising influencers on the outcome of Non-Alcoholic Fatty Liver Disease. In this review, The natural history of the evolution of NAFLD/NASH and the scarcity, if not total absence, of specific treatments for this condition necessitates increased attention to the possible application of innovative T2DM therapies also for this disease. Additionally, the citation and discussion of literature appear to be done in a thorough manner. I highly recommend its publication after a minor revision with the following comments addressed.

1. Please improve the resolution of the figure.

2. There are a few areas where the English could be improved, such as some past and present tense.

3. The CONCLUSION is perhaps the weakest part of the review - it is a little generic. It would be nice to have a few more thoughts from the authors about challenges and opportunities in this area for the future.

Author Response

We thank the referee for the valid suggestions according to which we changed the resolution of figure 1 (to 300 dpi) that has been uploaded (point 1).

As regard point 2. please find in attachment the receipt of the English revision process (by prof. Richard Burket).

Point 3. According to the suggestion of the referee the sentence of the Conclusion section "The natural history of the evolution of NAFLD/NASH and the scarcity, if not total absence, of specific treatments for this condition necessitate increased attention to the possible application of innovative T2DM therapies also for this disease. Of particular interest are SGLT2 inhibitors that efficiently reduce fat mass and prevent the release of inflammatory cytokines and, consequently, the inflammatory effects which are one of the main causes of NASH progression. Several studies have already been conducted on this class of drugs, making it one of the most promising future treatments for the specific indication of NAFLD"

has been changed as follows: 

"The natural history of the evolution of NAFLD/NASH and the scarcity, if not total absence, of specific treatments for this condition necessitate increased attention to the possible application of innovative T2DM therapies also for this disease. Of particular interest are SGLT2 inhibitors that have been proven to be efficient in reducing liver fat content, AST/ALT levels and even liver stiffness in several trials, making this class of drugs one of the most promising future treatments for the specific indication of NAFLD. Anyway most of the studies conducted so far are retrospective or conducted on small number of patients, therefore  prospective, randomized, controlled studies are needed to deepen the impact of SGLT2 inhibitors both on the already known clinical-instrumental parameters and on newer genetic and epigenetic predictors of NAFLD evolution".

Reviewer 2 Report

Dear Authors, I sincerely appreciate to your kind invitation me as a reviewer of recent manuscript, ijms-1651188, 'SGLT2 inhibitors as the most promising influencers on the outcome of Non-Alcoholic Fatty Liver Disease'.

This manuscript is well designed to review for pathogenic factors of NAFLD and put approaches of therapeutic, especially focusing on the SGLT-2. 

Regarding the explanation of NAFLD and trying to put the understanding of therapeutic accesses against NAFLD using SGLT-2 is very meaningful.

Thank you very much for your meaningful study and it would be very desirable to accept after minor revision of English grammatical errors and give a one figure of summary and brief diagram of your study as a figure 2.   

Reviewer was very happy to have a chance a great manuscript.

Thank you. 

Author Response

We thank for the revision made by the referee that suggested that: the manuscript is well designed to review for pathogenic factors of NAFLD and put approaches of therapeutic, especially focusing on the SGLT-2; regarding the explanation of NAFLD and trying to put the understanding of therapeutic accesses against NAFLD using SGLT-2 is very meaningful; and that therefore would be acceptable and to fit for the interest of journal purpose.